# Relative Training Load and Match Outcome: Are Professional Soccer Players Actually Undertrained during the In-Season?

**DOI:** 10.3390/sports9100139

**Published:** 2021-10-08

**Authors:** Toni Modric, Mario Jelicic, Damir Sekulic

**Affiliations:** 1Faculty of Kinesiology, University of Split, 21000 Split, Croatia; toni.modric@kifst.hr (T.M.); mario.jelicic@kifst.hr (M.J.); 2Football Club HNK Hajduk Split, 21000 Split, Croatia

**Keywords:** football, monitoring, external load, match running performance, success

## Abstract

Previous studies examined training/match ratios (TMr) to determine the training load relative to the match load, but the influence of the relative training load (RTL) on success in soccer is still unknown. Therefore, this study aimed to investigate the possible influence of RTL on final match outcome in soccer (win, draw, and loss). Running performances (RP) of soccer players (n = 21) in the Croatian highest national soccer competition were analyzed during the season 2020–2021. Data were measured by the global positioning system in 14 official matches and 67 training sessions. RTL was assessed by TMr, which were calculated as the ratio of RP during training and match in the same week, evaluating the following measures: TDr (total distance ratio), LIDr (low-intensity distance ratio), RDr (running distance ratio), HIDr (high-intensity distance ratio), ACCr (total accelerations ratio), DECr (total decelerations ratio), HI-ACCr (high-intensity accelerations ratio), HI-DECr (high-intensity decelerations ratio). All TMr were examined separately for each training session within in-season microcycles (categorized as days before the match day, i.e., MD minus). Spearman correlations were used to identify association between match outcome and TMr. The results indicated negative associations between match outcome and TDr, LIDr, ACCr and DECr on MD-1 and MD-2). In contrast, positive associations were evidenced between match outcome, and HIDr on MD-3 and TDr, LIDr, ACCr and DECr on MD-5 (*p* < 0.05; all moderate correlations). These findings demonstrate that final match outcome in soccer was associated with greater RTL of (i) high-intensity running three days before the match, (ii) total and low-intensity running, accelerations and decelerations five days before the match, and (iii) lower RTL of total and low-intensity running, accelerations and decelerations one and two days before the match.

## 1. Introduction

Longitudinal analysis of running performance (RP) in soccer indicates that the intensity level of soccer matches has increased tremendously over the last decade [1,2,3]. Specifically, Bernes et al. analyzed match RP between the 2006 and 2007 and 2012 and 2013 seasons in English Premier League soccer players and revealed that high-intensity running distance (>19.8 km/h) increased by approximately 30% in the 2012–2013 season [2]. Similar changes were noted also in other competitions [4]. To successfully cope with such increased match loads, the physical conditioning of the players has become an indispensable component of soccer training programs [5].

Basically, training programs are designed to enable players to achieve an appropriate conditioning status, prevent injuries, and compete at the highest possible level during the season [6]. To be effective, training programs in soccer must be related to the loads imposed during the matches (i.e., the relative training load (RTL)) [7]. However, due to a crowded match schedule of elite soccer teams (i.e., elite soccer teams may play one to three matches per week [8,9]), an appropriate periodization of the weekly microcycles may be challenging for the coaching staff. Consequently, in-season training programs are typically focused on the recovery from high-intensity match loads by utilizing passing drills, shooting practice, or small-sided games (SSG) [7,10,11,12]. Logically, such a training approach may help to avoid the overtraining of the players, while at the same time positively influencing technical and tactical aspects. Moreover, Aquire et al. in their review study highlighted that training approaches which utilize SSG may also improve the level of physical fitness [13]. However, since such training approaches usually contain drills that are performed in small areas, players are limited in reaching higher running speeds. As a consequence of not being exposed to high-intensity running patterns, players mostly do not meet the loads imposed during matches. In such cases, RTLs are very low, despite possibly intensive performed sessions (i.e., sessions with a high internal load). This may ruin the optimal physical conditioning of the players and even increase the risk of injury [14].

The quantification of training/match ratios is useful to understand how much players are exposed to match loads during the training sessions (i.e., the RTL) [14,15,16]. Since this method utilizes individual loads from training and matches, it also represents an important procedure for adjusting individual training programs according to the match loads [14,17,18]. This was well recognized over the last years and, consequently, has raised a growing interest in training/match ratios.

For example, Clemente et al. compared the accumulated weekly training load with the match load in the same week of Portuguese first division soccer players and reported the lowest training/match ratios for high-speed running and sprinting distance [14]. Stevens et al. quantified and compared the loads of Dutch first division soccer players between team’s training days and matches and reported the lowest training match ratios for high-speed running distance as well. Most specifically, this study indicated that the ratios for all analyzed RP were considerably lower than the match values in all training sessions during the week [15]. Additionally, Modric et al. recently analyzed the accumulated weekly training load and match loads in Croatian first division soccer players and reported higher ratios for total and low-intensity distance covered as well as for acceleration rates (ratios from 1.87 to 2.09), while the lowest ratio (0.92) was found for the high-intensity distance covered (>19.8 km/h) [16].

All previously cited studies clearly highlighted undertraining for high-intensity running in compared to the other running performances during the in-season weekly training practice of elite soccer teams in different countries. Although these findings provided valuable information for optimizing the physical conditioning of players, it is still unknown if such lower loads reflect a team’s success in matches. Given the fact that previous works demonstrated that running performance may affect key match activities (i.e., which are crucial for final match result) [19,20,21], it seems reasonable to investigate the possible influence of RTL (assed as training match/ratios) on success in soccer matches. Results of such investigation may enable the identification of crucial periods (i.e., days) in the week preceding the match which most significantly contribute to the team success in the subsequent match. Therefore, the main objectives of this study were to: (i) describe the daily training/match ratios of different RP variables in order to analyze intra-week variations of RTL, (ii) identify associations between training/match ratios and match outcome (win, draw, and loss) in order to investigate the possible influence of RTL on success in soccer. The analyses were conducted specifically for each training day during the week, as suggested previously by Clemente et al. [14]. Since high-intensity activities are important elements of success in soccer [22], the authors hypothesize that higher RTL (i.e., higher training/match ratios) for high-intensity running in the middle of the week will be associated with winning in the subsequent matches.

## 2. Materials and Methods

### 2.1. Participants

Twenty-one professional soccer players (M ± SD, age 24.19 ± 2.46, body mass 77.32 ± 4.45, height 182.32 ± 6.32) from the same team participated in this study. Players were classified according to the playing positions in the matches. Six of them were central midfielders (CM), four were central defenders (CD), four were forwards (FW), three were fullbacks (FB), and two were wide midfielders (WM). Goalkeepers were excluded due to the specificity of their role. 

All the players approved the use of training and match data for the purpose of the present study by written consent. The investigation was approved by the Ethical Board of the Faculty of Kinesiology, University of Split (approval number: 2181-205-02-05-19-0020, 1 September 2019)

### 2.2. Design

An observational cohort study was implemented on a professional soccer team during a full 2020/2021 season of the Croatian highest national soccer competition. All data were collected with the global positioning system (GPS) technique (see later for details) during in-season (i.e., competitive phase of the season) trainings and matches. We considered only those weeks in which three criteria were fulfilled: (i) the team played only one match in the week, (ii) there was minimum of 6 days between the matches, (iii) there was a minimum of 4 training days in the week. This decision was made to reduce the variability among comparisons, as previously suggested [14]. Due the well-known influence of situational and environmental factors on running performance in soccer [23,24], we also did not analyze matches that included red cards or specific outputs (e.g., bad weather, bad pitch, matches against teams that mathematically assured title or relegation) to minimize the influence of those contextual variables on RP. In addition, due to methodological reasons, only players who played a whole match and participated in all training sessions in the week before each match were included in the study. These criteria reduced number of the analyzed matches from 36 (i.e., which is maximum number of matches in observed competition) to 14. Consequently, only training sessions that preceded analyzed matches were observed (*n* = 67), resulting in 87 observations which were used as cases for this study. In the observed period, the team achieved six wins, three draws, and five losses and at the end of the season finished in the 4th position at the table (i.e., off 10 teams in total).

Training sessions were classified based on the number of days before the match day (i.e., MD minus) [15]. For example, MD-2 means that this session took place 2 days before the match day. From all analyzed weeks, 6 weeks consisted of five field-based training sessions, and 8 weeks of four field-based training days. The main differences between these two types of weeks were in the first 2 days after the match. Specifically, in weeks with five field-based training sessions, the first day after the match was always a day off, while on the second day after the match (i.e., MD-5), the team participated in an aerobic and technical session (i.e., the non-starter players had a “compensating” session immediately after the match). In weeks with four field-based training sessions, the first day after the match was a recovery session for the players that started the match and a “compensating” session for the players that did not start the match. The second day after the match (i.e., MD-5) in these weeks was always a day off. In both types of weeks, on MD-4, MD-3, MD-2, and MD-1, coaches applied similar training approaches. Specifically, MD-4 sessions commonly consisted of two training sessions: (i) a morning gym-based strength/power session, (ii) an afternoon soccer session based on technical–tactical drills and small-sided games. MD-3 sessions mostly included technical–tactical drills, small-sided games, large-sided games, and running drills. MD-2 and MD-1 were used for technical–tactical preparation for the subsequent match, applying tapering strategies (Figure 1).

### 2.3. Procedures

Apart from players’ age, body height, and body mass, variables in this study included two sets of RP variables, the final match outcome (observed as loss, draw, win), and the training/match ratio (TMr) for each RP variable as a measure of RTL.

Data on the RP of the players were collected by 21 GPS devices (Vector S7, Catapult, Catapult Sports Ltd., Melbourne) with a sampling frequency of 10 Hz. The use of this tracking system has appeared in previous researches [25,26]. Such device was already investigated for metrics, and was found to be appropriately reliable and valid in sport settings (i.e., less than 1% measurement error, and 80% of common variance with running speed measured by timing gates) [19,27,28]. Each player wore the same GPS device in all training sessions and matches in order to avoid inter-unit variability.

RP variables were measured (i) during the official matches (RMs), (ii) during training in the preceding week (RTs). The RMs were measured during official matches, as previously suggested [14]. The RTs were measured during all training sessions that the team participated in during the week and quantified according to the days that preceded official matches [15]. Both RMs and RTs included the total distance covered (m), low-intensity running (<14.3 km/h), running (14.4–19.7 km/h), high-intensity running (>19.8 km/h), high-speed running (19.8–25.1 km/h), sprinting (>25.2 km/h), number (frequency) of total accelerations (>0.5 m/s^2^), number of high-intensity accelerations (>3 m/s^2^); number of total decelerations (less than −0.5 m/s^2^), and number of high-intensity decelerations (less than –3 m/s^2^) [15].

RTL was assessed by TMr, by dividing the training demands (i.e., RTs) by the match demands (i.e., RMs) based on the following formula: TMr = RT/RM [14]. TMr was calculated for each RP variable, separately on each training day. Therefore, the following TMr values were examined: TDr (total distance ratio), LIDr (low-intensity distance ratio), RDr (running distance ratio), HIDr (high intensity distance ratio), ACCr (total accelerations ratio), DECr (total decelerations ratio), HI-ACCr (high-intensity accelerations ratio), HI-DECr (high-intensity decelerations ratio). All TMr are expressed in arbitrary units (AU).

### 2.4. Statistics

Homogeneity was checked by the Levene’s test. The normality of the distributions was confirmed by the Kolmogorov–Smirnov test, and data are presented as means ± standard deviations. All variables were normally distributed (all K-S *p* > 0.05), except TDr-MD4, LIDr-MD4 and HIDr-MD4 (K-S: *p* < 0.05, 0.05 and 0.01, respectively). Therefore, later statistical analyses were adapted accordingly (please see later for details. 

The differences in TDr, LIDr and HIDr among training days were analysed by non-parametric Friedman’s test, while differences in all other TMr among training days were analysed by repeated-measures ANOVA. 

One-way analysis of variance was used to examine differences in TMr according to match outcome for normally distributed variables. On the other hand, differences in TDr-MD4, LIDr-MD4 and HIDr-MD4 according to the match outcome were examined by non-parametric Kruskal-Wallis test. 

Spearman’s correlations were used to identify the association between match outcome and TMr. For such purpose matches were coded with numbers (1 = loss, 2 = draw, 3 = win), and correlations were classified as previously suggested: 0 = no correlation, 0 < |r| < 0.2 = very weak correlation, 0.2 ≤ |r| < 0.4 = weak correlation, 0.4 ≤ |r| < 0.6 = moderate correlation, 0.6 ≤ |r| < 0.8 = strong correlation, 0.8 ≤ |r| < 1.0 = very strong correlation, and 1 = perfect correlation [29]. Due to the multiple correlations (n = 40), we used the Bonferroni method for family-wise error rate correction, and adjusted significance level of *p* ≤ 0.013 was applied [30].

Scheffe post-hoc test was used to examine the specific differences among: (i) match outcomes, and (ii) training days for normally distributed variables, while Mann-Whitney U test was applied to examine specific differences for TDr-MD4, LIDr-MD4 and HIDr-MD4. 

Hedges’g effect sizes were calculated to determine the magnitude of differences in TMr between specific match outcomes (e.g., loss to draw, draw to win, win to loss) for parametric variables, and classified as: trivial (<0.2), small (0.2–0.5), moderate (0.5–0.8), and large (>0.8) [31]. Results of Mann-Whitney-U tests were converted in effect sizes for non-parametric variables [32]. 

For all analyses, Statistica (Version 16; TIBCO Software, Palo Alto, CA, USA) was used, and a significance level α = 0.05 was applied.

## 3. Results

Figure 2 presents descriptive statistics for training/match ratios of the variables of running performances. Additionally, ANOVA differences among training days are noted within each graph for each variable. In brief, significant ANOVA differences among training sessions were found for all TMr (all *p* < 0.01). Specifically, TDr and LIDr were highest on MD-3 (0.62 and 0.65, respectively) (significant post hoc differences when compared to MD-1 (0.38 and 0.42, respectively), MD-2 (0.4 and 0.46, respectively) and MD-4 (0.55 and 0.58, respectively)). Highest RDr was evidenced on MD-5 (0.57) (significant post hoc differences when compared to MD-1 (0.25), MD-2 (0.23) and MD-4 (0.44)), while highest HIDr was found on MD-3 (0.56) (significant post hoc differences when compared to all MD minus days (MD-1,2,4 and 5: 0.19, 0.15, 0.48 and 0.35, respectively)). The lowest values of TDr, LIDr, RDr and HIDr were evidenced on MD-1 (0.38, 0.42, 0.25 and 0.19, respectively) and MD-2 (0.4, 0.46, 0.23 and 0.15, respectively) (all significant post hoc difference when compared to MD-3).

Descriptive statistics (means ± standard deviations) for the ratios of the acceleration/deceleration variables are presented in Figure 3, together with ANOVA results (differences among training days). The ACCr and DECr were highest on MD-3 (both 0.63) (significant post hoc differences when compared to MD-1 (both 0.36), MD-2 (0.36 and 0.37, respectively) and MD-5 (0.51 and 0.52, respectively)). Highest values of HI-ACCr and HI-DECr were also evidenced on MD-3 (0.96 and 0.59, respectively) (significant post hoc differences when compared to MD-1 (0.41 and 0.25, respectively) and MD-2 (0.37 and 0.23, respectively)). ACCr, DECr, HI-ACCr and HI-DECr were lowest on MD-1 (0.36, 0.36, 0.41 and 0.25, respectively) and MD-2 (0.36, 0.37, 0.37, and 0.23, respectively) (significant post hoc difference when compared to MD-3 (0.63, 0.63, 0.96, and 0.59, respectively) and MD-4 (0.55, 0.56, 0.87, and 0.59, respectively)).

When team won, significantly lowest values of TDr and LIDr were evidenced in preceding weeks on MD-1 (0.35 and 0.39, respectively) and MD-2 (0.35 and 0.4, respectively) (F-test: from 10.37 to 16.98; all *p* < 0.01). On the other hand, results indicated significantly highest values of TDr and LIDr on MD-5 (0.71 and 0.72, respectively; F-test: 31.65 and 34.40, respectively; both *p* < 0.01) and highest values of HIDr on MD-3 (0.73; F-test: 7.69; *p* < 0.01) if team won in subsequent match (Table 1).

Significantly lower values of ACCr and DECr were evidenced on MD-1 (0.32 and 0.33, respectively) and MD-2 (0.29 and 0.3, respectively) (F-test: from 14.51 to 16.72; all *p* < 0.01) when team won in subsequent matches. In addition, results indicated significantly higher values of ACCr and DECr on MD-5 (0.65 and 0.66, respectively; F-test: 10.76 and 12.43, respectively; both *p* < 0.01) in weeks that preceded won matches (Table 2). 

Significant negative associations between match outcome and TDr, LIDr, ACCr and DECr were evidenced on MD-1 and MD-2 (*p* < 0.001; all moderate correlations). In contrast, positive association between match outcome and HIDr was evidenced on MD-3 (*p* < 0.001; moderate correlation). Additionally, results indicated positive association between match outcome and TDr, LIDr, ACCr and DECr on MD-5 (*p* < 0.001; all moderate correlations) (Table 3).

## 4. Discussion

This study provides several important findings. First, RTL (assessed by TMr) were the highest on MD-3 and the lowest on MD-1 and MD-2. Second, accelerations, decelerations, total, and low-intensity distance covered during trainings were, in general, closest to the corresponding match values, indicating that the observed players were undertrained in high-intensity running during in-season training sessions. Third, the identified associations between match outcome and all TMr indicate an important influence of RTL on success in soccer.

### 4.1. Relative Training Load within In-Season Microcycles

Our results evidenced significant differences for all TMr within weekly microcycles (F-test: 10.10 to 16.65; all *p* < 0.05; all large ES), indicating significant variability of RTL on different training days. Specifically, all TMr were the highest on MD-3 and the lowest on MD-1 and MD-2. In other words, RPs in training sessions (i.e., RTs) were closest to the RP in matches (i.e., RMs) on MD-3, while the training load in general decreased as the match day approached. Such findings are partly in line with previous studies which investigated the distribution of the weekly training load in English and Dutch soccer players [9,15,33]. Briefly, Akenhead et al., Anderson et al., and Stevens et al. reported the lowest training load on MD-1 and the highest training load on MD-3 and MD-4, showing differences in the distribution of the training load in professional soccer players [15]. These inconsistencies in findings are most likely influenced by different training methodologies in different countries.

Comparison of values of different TMr may provide insights for a better understanding of a training structure. For example, previous studies provided clear evidence that acceleration rate ratios are greater than those for distance covered at high speed (>19.8 km/) [14,15]. In agreement with this, our study evidenced the highest TMr for HI-ACCr (0.97), followed by LIDr (0.65), TDr (0.63), ACCr (0.63), DECr (0.63), and HI-DECr (0.59), while the lowest TMr were observed for RDr and HIDr (0.56 and 0.53, respectively). Such results provide three important findings.

First, players were not generally exposed to the RP that are normally observed during the matches, even in the training session with the highest overall training load (MD-3). Second, in the highest physically demanding training sessions, players averagely performed 97% of high-intensity accelerations and reached approximately 65% of total and low distance covered, total accelerations, and total and high decelerations and only ~55% of distance covered at high and moderate speed with respect to matches. Third, players were evidently undertrained for high-intensity running in the highest physically demanding training session.

As described previously, such training structure is typical of sessions where small-sided games are preferred and utilized [14,15]. In detail, these games increase the frequency of accelerations/decelerations while decreasing opportunities to perform high-speed running (19.8–25.1 km/h) or sprinting (>25.2 km/h). Since repeated-sprint ability [7,34] and high-intensity activities [19,22] are critical aspects in soccer, the authors of this study believe that players should be exposed to higher RTL of high-intensity running (>19.8 km/h) during the training sessions with the highest training loads (i.e., MD-3 in this study). Basically, ensuring appropriate bouts of high-intensity activities in in-season training sessions will maintain players’ fitness at the optimum level [7].

### 4.2. Association between Relative Training Load and Success in Soccer

Results from our study evidenced positive association between match outcome and HIDr on MD-3, indicating greater RTL of high-intensity running three days before winning matches (*p* < 0.001; moderate correlation). Thus, HIDr on MD-3 were significantly higher when the team won (0.73), than when it drew or lost (0.46 and 0.37, respectively) (F-test: 7.69; *p* < 0.01). Specifically, the RTL of high-intensity running on MD-3 was 50% higher when the team won than when it lost in the subsequent match. Knowing the importance of high-intensity running in soccer, such associations are not surprising. In general, many key elements that are crucial for the final match result in soccer (i.e., pressing, attacking the space, counterattacking, defensive transition, etc.), are affected by high-intensity running [20,21]. On the other hand, high-intensity running directly corresponds to the intensity of the anaerobic threshold [35], and a higher RTL actually improves the anaerobic capacities, leading to better adaptation during high-intensity running that occurs in the match (i.e., when appropriate periodization is applied). Logically, with continuous weekly exposure to high RTL, elements that are affected by high-intensity running may be more efficiently performed in the matches, which can ultimately lead to a positive final result. 

Furthermore, we evidenced positive associations between match outcome and TDr, LIDar, AACr, and DECr on MD-5 (*p* < 0.001; all moderate correlations). Indeed, significantly higher TDr, LIDar, AACr, and DECr were found on MD-5 when the team won than when lost in the subsequent match. In other words, MD-5 training sessions were characterized by higher RTL of total and low intensity running, accelerations and decelerations when the team won in the same week. Since these RPs typically determine the volume of the trainings/matches [36], it seems that a higher volume of training session conducted on MD-5 may contribute to achieving success in the next match. 

Authors of this study were deeply involved in the training and conditioning of the players examined in this study. Consequently, we are convinced that the reason for these findings (e.g., higher TDr, LIDar, AACr, and DECr on MD-5 when team won) directly corresponds to the applied training methodology in weeks with five training days. Specifically, when the team had a day off the first day after the match, a typical MD-5 training session aimed to improve aerobic capacity and technical skills (since recovery from the match was still incomplete). Such sessions included low- and moderate-speed running drills or low-demanding soccer drills that required total, low-, and moderate-intensity running, accelerations, and decelerations. Given the fact that these activities are related to the intensity of the aerobic threshold [37,38], constantly higher RTL of these variables (i.e., greater TDr, LIDr, ACCr and DECr on MD-5) probably allowed players to achieve a better adaptation of their aerobic capacities. Considering the fact that poor endurance preparation of players results in a rapid increase in fatigue, a greater number of errors, and more frequent loss of one-on-one plays [39], a better adaptation of the aerobic capacities during training will probably result in the ability to manage prolonged fatigue during the matches. Logically, it may allow players to efficiently execute technical–tactical actions, which may even affect the final match result. 

It is well known that a decrease in the training load after heavy sessions which are characterized by high internal and external load is required for optimizing players’ performance on the match day (i.e., tapering strategy) [40,41,42]. In support to this, we evidenced negative association between match outcome and that TDr, LIDar, ACCr, and DECr on MD-1 and MD-2 (*p* < 0.001; all moderate correlations). Indeed, TDr, LIDar, ACCr, and DECr on MD-1 and MD-2 were significantly lower when the team won in the subsequent match than when it drew or lost (all *p* < 0.01). In other words, RTL for total and low-intensity running and total accelerations and decelerations were lower by approximately 20–25% on both days before the match when the team won in the next match than when it drew or lost. These findings are actually in agreement with what previously discussed, indicating the importance of applying a tapering strategy two days before the match in order to maximize players’ performance. Most probably, reducing the training volume (i.e., total and low-intensity running, total accelerations and decelerations) before the matches may decrease the physiological and psychological stress of training and maximize performance after an intense training period [40,41,42]. Consequently, it may positively influence the final result in the subsequent match. 

### 4.3. Strengths and Limitations

The main limitation of this study is the fact that only one team was observed. However, this is a very common obstacle in studies involving professional and elite players [15,43]. In addition, we included players only if they participated in full training sessions and the corresponding following matches.

To the best of knowledge, this is the first study that analyzed associations between success in soccer and RTL in the preceding weeks, clustered by training days. Further, this study provides novel knowledge for soccer coaches and strength and conditioning staff, detailing the dissimilar running demands of in-season field-based training and its influence on matches’ final results. In addition, our findings strongly emphasize the importance of proper training in high-intensity running during the in-season, reinforcing previous evidence that clearly highlighted players’ undertraining in in-season high intensity-running.

### 4.4. Practical Implications and Future Research

Players should be exposed to a minimum of 75–80% of the high-intensity running normally characterizing a match in the middle of the week (i.e., on MD-3 in our study);When utilizing small-sided games which do not require running at high speeds, coaches should include specific running drills that entail high-intensity running (e.g., high-speed running and sprinting) in the training sessions;Training methodology that utilizes a “high-volume and low-intensity” training session the second day after the match (i.e., on MD-5) may positively impact success in soccer;After heavy training sessions which are characterized by high internal and external load in the middle of the week, coaches should reduce the volume of the training on the two days before the match.For a more comprehensive understanding of RTL, measures of internal load should be included in future studies.

## 5. Conclusions

The results from our study demonstrate the influence of RTL within weekly training sessions on success in soccer. Specifically, we found that a final match result in soccer may be influenced by a greater RTL of performances that determine the training intensity (i.e., high-intensity running) three days before the match.

Additionally, this study indicated that a greater RTL of variables that determine the training volume (i.e., total and low-intensity running, accelerations and decelerations) 2 days after the match (i.e., on MD-5) may positively reflect on a match final result.

Finally, we found that a decrease in RTL variables that determine the training volume 1 and 2 days before a match increases the possibility of winning the match. These findings indicate that such approach will maximize players’ performance on match day and, at the same time, reinforce previous knowledge about the effectiveness of tapering strategies in soccer.

## Figures and Tables

**Figure 1 sports-09-00139-f001:**
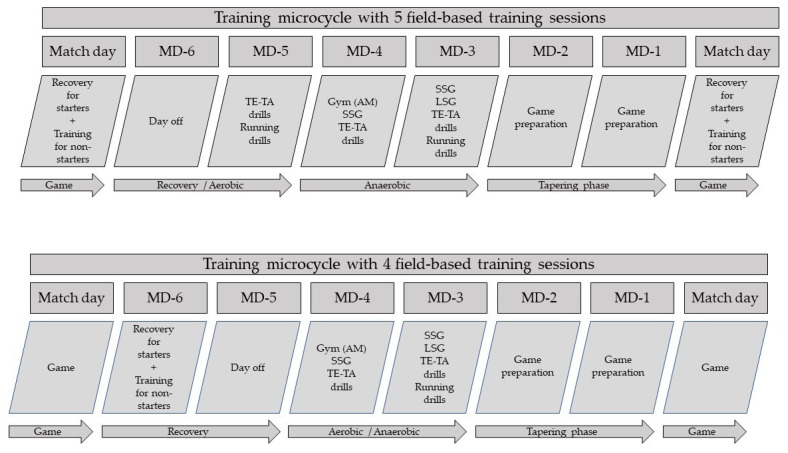
Differences between microcycles with five and four field-based training sessions (TE-TA = technical tactical, SSG = small sided games, LSG = large sided games, MD = Match day).

**Figure 2 sports-09-00139-f002:**
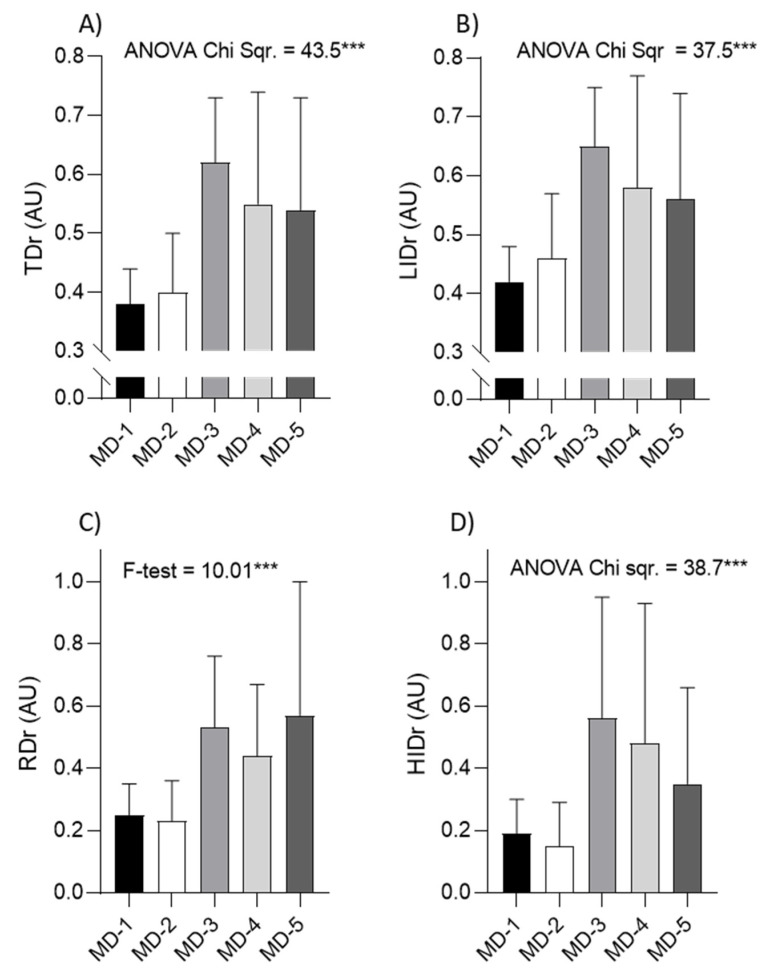
Descriptive statistics (means ± standard deviations), and results of the parametric (F-test), or non-parametric (ANOVA Chi sqr.) analysis of variance (*** *p* < 0.001) among training days (MD-1, MD-2, MD-3, MD-4, MD-5) for total distance ratio (TDr; (**A**)), low-intensity distance ratio (LIDr; (**B**)), running distance ratio (RDr; (**C**)) and high intensity distance ratio (HIDr; (**D**)).

**Figure 3 sports-09-00139-f003:**
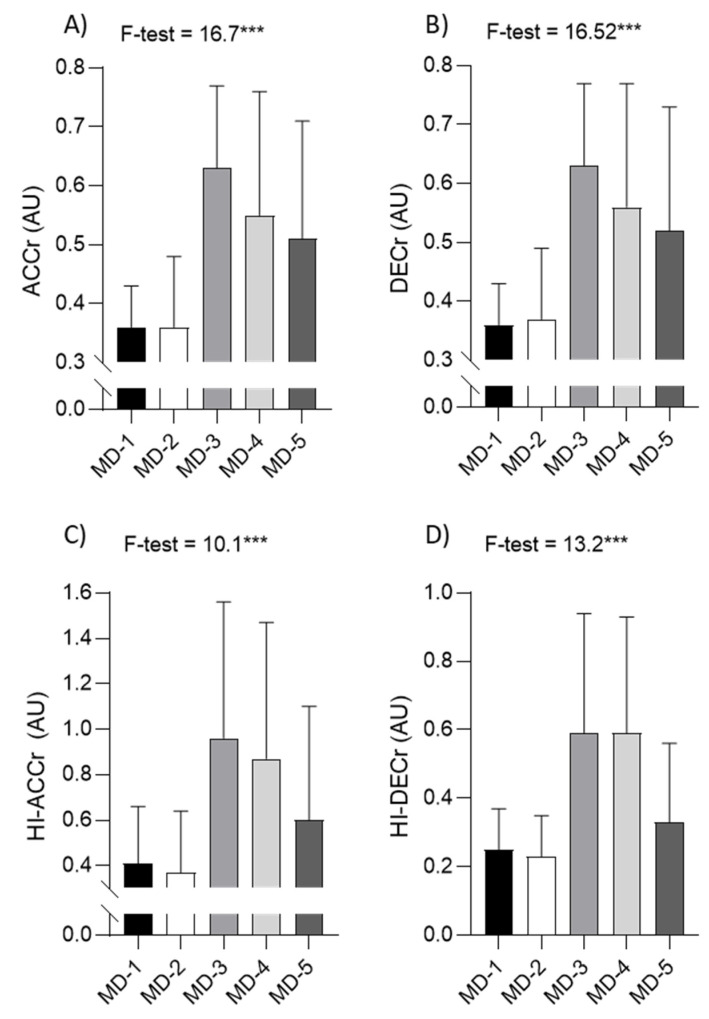
Descriptive statistics (means ± standard deviations), and results of the analysis of the variance (F-test, *** *p* < 0.001) among training days (MD-1, MD-2, MD-3, MD-4, MD-5) for total accelerations ratio (ACCr; (**A**)), total accelerations ratio (DECr; (**B**)), high intensity accelerations ratio (HI-ACCr; (**C**)) and high intensity decelerations ratio (HI-DECr; (**D**)).

**Table 1 sports-09-00139-t001:** Differences in total distance ratio (TDr), low-intensity distance ratio (LIDr), running distance ratio (RDr) and high intensity distance ratio (HIDr) according to the match outcome (data are given as mean ± SD).

		Match Outcome			Analysis of Variance		Effect Size		
		Loss	Draw	Win	F-Test/ H-Test *	*p*	L/W	D/L	D/W
TDr	MD-1	0.42 ± 0.04 ^W^	0.4 ± 0.05 ^W^	0.35 ± 0.05 ^L, D^	16.98	0.01	1.41	0.43	0.95
	MD-2	0.44 ± 0.12 ^W^	0.45 ± 0.06 ^W^	0.35 ± 0.09 ^L, D^	10.37	0.01	0.89	0.07	1.21
	MD-3	0.59 ± 0.06	0.61 ± 0.11	0.63 ± 0.12	1.00	0.37	0.40	0.26	0.15
	MD-4 *	0.48 ± 0.1 ^D^	0.73 ± 0.22 ^L, W^	0.48 ± 0.14 ^D^	21.08	0.01	0.03	1.53	1.29
	MD-5	0.43 ± 0.11 ^W^	0.36 ± 0.04 ^W^	0.71 ± 0.13 ^L, D^	31.65	0.01	2.12	0.91	3.05
LIDr	MD-1	0.46 ± 0.04 ^W^	0.44 ± 0.06 ^W^	0.39 ± 0.05 ^L, D^	12.50	0.01	1.28	0.39	0.78
	MD-2	0.49 ± 0.12 ^W^	0.5 ± 0.07 ^W^	0.4 ± 0.09 ^L, D^	10.47	0.01	0.89	0.11	1.24
	MD-3	0.63 ± 0.06	0.66 ± 0.09	0.65 ± 0.11	0.46	0.64	0.18	0.35	0.09
	MD-4 *	0.5 ± 0.13 ^D^	0.79 ± 0.2 ^L, W^	0.51 ± 0.13 ^D^	30.09	0.01	0.02	1.99	1.78
	MD-5	0.43 ± 0.16 ^W^	0.41 ± 0.04 ^W^	0.72 ± 0.08 ^L, D^	34.40	0.01	2.50	0.12	4.37
RDr	MD-1	0.26 ± 0.1	0.28 ± 0.08	0.22 ± 0.1	2.58	0.08	0.35	0.24	0.61
	MD-2	0.25 ± 0.14	0.27 ± 0.1	0.18 ± 0.13	3.73	0.03	0.52	0.10	0.72
	MD-3	0.46 ± 0.11	0.49 ± 0.22	0.59 ± 0.27	2.65	0.08	0.59	0.20	0.39
	MD-4	0.4 ± 0.14 ^D^	0.59 ± 0.32 ^L, W^	0.37 ± 0.18 ^D^	7.19	0.01	0.23	0.77	0.89
	MD-5	0.59 ± 0.16	0.18 ± 0.05 ^W^	0.78 ± 0.51 ^D^	6.97	0.01	0.43	3.35	1.40
HIDr	MD-1	0.21 ± 0.1	0.23 ± 0.11	0.15 ± 0.11	4.40	0.02	0.53	0.24	0.72
	MD-2	0.16 ± 0.17	0.22 ± 0.15	0.1 ± 0.11	4.48	0.01	0.43	0.34	0.90
	MD-3	0.37 ± 0.2 ^W^	0.46 ± 0.24 ^W^	0.73 ± 0.47 ^L, D^	7.69	0.01	0.91	0.40	0.68
	MD-4 *	0.43 ± 0.42	0.61 ± 0.6	0.45 ± 0.51	0.97	0.62	0.13	0.23	0.23
	MD-5	0.3 ± 0.2	0.1 ± 0.04	0.52 ± 0.47	4.02	0.03	0.53	1.33	1.08

Superscripted letters indicate significant differences in TMr (^W^ = significantly different from won matches; ^D^ = significantly different from drawn matches; ^L^ = significantly different from lost matches). L/W—effect size between loss and win, D/L—effect size between draw and loss, D/W—effect size between draw and win; * denotes variables where Kruskal Wallis ANOVA was calculated.

**Table 2 sports-09-00139-t002:** Differences in total accelerations ratio (ACCr), total accelerations ratio (DECr), high intensity accelerations ratio (HI-ACCr) and high intensity decelerations ratio HI-DECr according to match outcome (data are given as mean ± SD).

		Match Outcome			Analysis of Variance		Effect Size		
		Loss	Draw	Win	F-Test	*p*	L/W	D/L	D/W
ACCr	MD-1	0.4 ± 0.05 ^W^	0.38 ± 0.06 ^W^	0.32 ± 0.06 ^L, D^	16.21	0.01	1.44	0.51	0.86
	MD-2	0.44 ± 0.1 ^W^	0.39 ± 0.08 ^W^	0.29 ± 0.11 ^L, D^	16.76	0.01	1.38	0.55	0.93
	MD-3	0.63 ± 0.08	0.64 ± 0.19	0.63 ± 0.13	0.03	0.97	0.04	0.03	0.06
	MD-4	0.49 ± 0.21 ^D^	0.74 ± 0.2 ^L, W^	0.48 ± 0.12 ^D^	15.82	0.01	0.07	1.18	1.60
	MD-5	0.43 ± 0.2 ^W^	0.34 ± 0.07 ^W^	0.65 ± 0.16 ^L, D^	10.76	0.01	1.16	0.56	2.11
DECr	MD-1	0.4 ± 0.05 ^W^	0.38 ± 0.06 ^W^	0.33 ± 0.06 ^L, D^	14.72	0.01	1.39	0.50	0.81
	MD-2	0.44 ± 0.1 ^W^	0.39 ± 0.09 ^W^	0.3 ± 0.12 ^L, D^	14.51	0.01	1.28	0.57	0.83
	MD-3	0.63 ± 0.09	0.63 ± 0.18	0.63 ± 0.13	0.02	0.98	0.06	0.05	0.00
	MD-4	0.49 ± 0.2 ^D^	0.75 ± 0.21 ^L, W^	0.49 ± 0.12 ^D^	17.48	0.01	0.03	1.26	1.62
	MD-5	0.44 ± 0.2 ^W^	0.34 ± 0.06 ^W^	0.66 ± 0.16 ^L, D^	12.43	0.01	1.19	0.69	2.29
HI-ACCr	MD-1	0.35 ± 0.16	0.44 ± 0.23	0.44 ± 0.31	1.03	0.36	0.32	0.44	0.01
	MD-2	0.36 ± 0.26	0.36 ± 0.26	0.38 ± 0.3	0.04	0.96	0.07	0.03	0.04
	MD-3	1.17 ± 0.81 ^D^	0.71 ± 0.32 ^L^	1 ± 0.55	3.43	0.04	0.26	0.75	0.60
	MD-4	0.54 ± 0.28 ^D, L^	1.07 ± 0.86 ^L^	1.01 ± 0.52 ^L^	6.66	0.01	1.07	0.86	0.09
	MD-5	0.31 ± 0.17 ^W^	0.27 ± 0.2 ^W^	0.91 ± 0.54 ^D, L^	8.59	0.01	1.26	0.20	1.37
HI-DECr	MD-1	0.24 ± 0.07	0.27 ± 0.12	0.25 ± 0.16	0.29	0.75	0.07	0.30	0.13
	MD-2	0.27 ± 0.12 ^W^	0.23 ± 0.09	0.19 ± 0.12 ^L^	4.13	0.02	0.69	0.37	0.39
	MD-3	0.71 ± 0.42	0.56 ± 0.27	0.55 ± 0.35	1.44	0.24	0.42	0.41	0.04
	MD-4	0.45 ± 0.27	0.68 ± 0.43	0.63 ± 0.31	3.26	0.04	0.60	0.64	0.12
	MD-5	0.22 ± 0.14	0.29 ± 0.14	0.42 ± 0.28	2.38	0.11	0.81	0.46	0.53

Superscripted letters indicate significant post-hoc differences in TMr (^W^ = significantly different from won matches; ^D^ = significantly different from drawn matches; ^L^ = significantly different from lost matches). L/W—effect size between loss and win, D/L—effect size between draw and loss, D/W—effect size between draw and win.

**Table 3 sports-09-00139-t003:** Spearman’s correlation between TMr and match outcome (win, draw, loss) (data are given as r (*p*)).

	MD-1	MD-2	MD-3	MD-4	MD-5
TDr	−0.54 (0.001) *	−0.40 (0.001) *	0.20 (0.09)	−0.01 (0.91)	0.72 (0.001) *
LIDr	−0.50 (0.001) *	−0.40 (0.001) *	0.07 (0.58)	−0.03 (0.80)	0.73 (0.001) *
RDr	−0.22 (0.04)	−0.25 (0.03)	0.2 (0.048)	−0.08 (0.48)	0.15 (0.42)
HIDr	−0.29 (0.001) *	−0.19 (0.08)	0.40 (0.001) *	−0.05 (0.67)	0.25 (0.19)
ACCr	−0.54 (0.001) *	−0.53 (0.001) *	0.00 (0.98)	−0.09 (0.42)	0.56 (0.001) *
DECr	−0.52 (0.001) *	−0.51 (0.001) *	0.06 (0.62)	−0.06 (0.60)	0.57 (0.001) *
HI-ACCr	0.06 (0.62)	0.02 (0.83)	0.00 (0.98)	0.35 (0.001) *	0.60 (0.001) *
HI-DECr	−0.09 (0.41)	−0.30 (0.001) *	−0.14 (0.24)	0.23 (0.04)	0.33 (0.07)

* denotes significant correlations at adjusted *p* < 0.013.

## Data Availability

Data will be provided to all interested parties upon reasonable request.

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
