# Peer review of "Relative Training Load and Match Outcome: Are Professional Soccer Players Actually Undertrained during the In-Season?"

_sports, 2021, doi:10.3390/sports9100139_

Round 1

Reviewer 1 Report

ABSTRACT

Lines 9-12: maybe can be re-written. It seems that the objective was to analyze the variations of TMr between match outcomes and not test relationships (that are more related to correlations).

Lines 12-14: authors must describe the number of matches analyzed, data used, and eligibility criteria.

INTRODUCTION

Lines 73-76: a doubt can result from this sentence. What is the weight of the match running on the final match outcome? Is still so obvious that running more or less will increase the odds of scoring more than the other team? A better rationale should be done since is the main topic of the current article. As example, this article does not confirm such an idea: Hoppe, M. W., Slomka, M., Baumgart, C., Weber, H., & Freiwald, J. (2015). Match running performance and success across a season in German Bundesliga soccer teams. International journal of sports medicine, 36(07), 563-566.

Additionally, this one reveals that match performance is highly complex and dependent on different variables: Geurkink, Y., Boone, J., Verstockt, S., & Bourgois, J. G. (2021). Machine Learning-Based Identification of the Strongest Predictive Variables of Winning and Losing in Belgian Professional Soccer. Applied Sciences, 11(5), 2378.

Lines 76-91: maybe is better to reorganize this paragraph, namely after improving the rationale. Additionally, a statement of contribution should be added before objectives.

METHODS

Line 98: add the eligibility criteria. Additionally, explain how much many of sessions per week were considered for the player being included in the analysis. How the authors managed the missing cases (example: not participating in all training sessions, but playing on the weekend).

Line 104: start with the type of study design.

Lines 143-146: describe how many GPS units were used. Each player used the same? How the variability was controlled if not?

Lines 145-146: add the reliability and validity levels for each of the main outcomes.

Lines 166-168: add the normality and homogeneity levels of each of the outcomes

Line 169: the repeated measures were made by each player? How did the authors manage the missings?

RESULTS

Figures: add the units of measurement in the y-axis

The statistical report in the text must describe the percentage of differences in the pairwise comparisons.

DISCUSSION

A sub-section of future research and practical implications should be added at the bottom of the section.

Author Response

REVIEWER 1

Thank you very much for all comments and suggestions. We tried to follow all of them and revised manuscript accordingly.

ABSTRACT

Lines 9-12: maybe can be re-written. It seems that the objective was to analyze the variations of TMr between match outcomes and not test relationships (that are more related to correlations).

Thank you for this suggestion. We re-wrote this part and text now reads: “Previous studies examined training/match ratios (TMr) to determine the training load relative to the match load, but the influence of the relative training load (RTL) on success in soccer is still unknown. Therefore, this study aimed to investigate the possible influence of RTL on final match outcome in soccer (win, draw, and loss).”

Lines 12-14: authors must describe the number of matches analyzed, data used, and eligibility criteria.

We amended this part according to your suggestion. Text now reads: “Running performances (RP) of soccer players (n = 21) in the Croatian highest national soccer competition were analyzed during the season 2020–2021. Data were measured by the global positioning system in 14 official matches and 67 training sessions”

INTRODUCTION

 Lines 73-76: a doubt can result from this sentence. What is the weight of the match running on the final match outcome? Is still so obvious that running more or less will increase the odds of scoring more than the other team? A better rationale should be done since is the main topic of the current article. As example, this article does not confirm such an idea: Hoppe, M. W., Slomka, M., Baumgart, C., Weber, H., & Freiwald, J. (2015). Match running performance and success across a season in German Bundesliga soccer teams. International journal of sports medicine, 36(07), 563-566.

Additionally, this one reveals that match performance is highly complex and dependent on different variables: Geurkink, Y., Boone, J., Verstockt, S., & Bourgois, J. G. (2021). Machine Learning-Based Identification of the Strongest Predictive Variables of Winning and Losing in Belgian Professional Soccer. Applied Sciences, 11(5), 2378.

Thank you for your comment. Regarding your questions “What is the weight of the match running on the final match outcome? Is still so obvious that running more or less will increase the odds of scoring more than the other team?  – first we need to say that in this part we did not refer on match running performance. We wrote: All previously cited studies clearly highlighted undertraining for high-intensity running during the inseason weekly training practice of elite soccer teams in different countries. Although these findings provided valuable information for optimizing the physical conditioning of players, it is still unknown if high-intensity training loads reflect a team’s success in matches. As you see, here we were writing about running performance at training sessions and possible influence of running performance during training on match outcome. The papers you cited are about match running performance, but not about training running performance.

However, we believe that key match activities, which are crucial for final match result, are actually affected by greater RP (please see: https://pubmed.ncbi.nlm.nih.gov/ 28488830; https://pubmed.ncbi.nlm.nih.gov/30455549). Briefly, to successfully attack the last third of the pitch (i.e., by forwards), to perform efficient counterattacks (i.e., by wide midfielder), to go in last third of the pith for crossing (i.e., by fullbacks) or to perform defensive transition (i.e., by central midfielders and central defenders), players should create spatiotemporal advantage over the opponent players. This can be done only by utilizing running at higher speeds and high intensity accelerations/decelerations when performing such activities. Logically, higher amount of such activities will provoke greater high-speed running, sprinting and high intensity accelerations/decelerations, and will consequently enable players more opportunities for scoring the goals. Finally, creating more of such situations in the match may provoke even higher overall distance covered, hypothesizing that greater RP may influence success in football.

Therefore, we thought that it’s reasonable to investigate the possible influence of RTL (assessed by training match/ratios) on success in soccer matches. Especially if consider the fact that previous studies highlighted high-intensity activities as really one of the crucial elements of success in soccer (please see: https://pubmed.ncbi.nlm.nih.gov/19214939/).

Finally, we rewrote this part and added two more references to emphasize importance of high intensity running in soccer.

Lines 76-91: maybe is better to reorganize this paragraph, namely after improving the rationale. Additionally, a statement of contribution should be added before objectives.

Thank you very much. We agree with your suggestion to reorganize this paragraph together with improving the rationale. We did it, and text now reads: “All previously cited studies clearly highlighted undertraining for high-intensity running during the in-season weekly training practice of elite soccer teams in different countries. Although these findings provided valuable information for optimizing the physical conditioning of players, it is still unknown if high-intensity training loads reflect a team’s success in matches. Given the fact that previous works demonstrated that running performance may affect key match activities (i.e., which are crucial for final match result) [19-21], it seems reasonable to investigate the possible influence of RTL (assed as training match/ratios) on success in soccer matches. Results of such investigation may enable the identification of crucial periods (i.e., days) in the week pre-ceding the match which most significantly contribute to the team success in the sub-sequent match. Therefore, the main objectives of this study were to: (i) describe the daily training/match ratios of different RP variables in order to analyze intra-week variations of RTL, (ii) identify associations between training/match ratios and match outcome (win, draw, and loss) in order to investigate the possible influence of RTL on success in soccer. The analyses were conducted specifically for each training day during the week, as suggested previously by Clemente et al. [14]. Since high-intensity activities are important elements of success in soccer [22], the authors hypothesize that higher RTL (i.e., higher training/match ratios) for high-intensity running in the middle of the week will be associated with winning in the subsequent matches.”

METHODS

Line 98: add the eligibility criteria. Additionally, explain how much many of sessions per week were considered for the player being included in the analysis. How the authors managed the missing cases (example: not participating in all training sessions, but playing on the weekend).

Thank you for your comment. We added explanation that players were classified according to their playing positions in the game. Additionally, we added information that players averagely participated in 4.4 field-based training sessions per week. Text now reads: “Twenty-one professional soccer players (M ± SD, age 24.19 ± 2.46, body mass 77.32 ± 4.45, height 182.32 ± 6.32) from the same team participated in this study. Players were classified according to the playing positions in the matches. Six of them were central midfielders (CM), four were central defenders (CD), four were forwards (FW), three were fullbacks (FB), and two were wide midfielders (WM). Goalkeepers were excluded due to the specificity of their role. Players averagely participated in 4.4 field-based training sessions per week.”

Line 104: start with the type of study design.

We amended this part according to your suggestion. Text now reads: “An observational cohort study was implemented on a professional soccer team during a full 2020/2021 season of the Croatian highest national soccer competition.”

Lines 143-146: describe how many GPS units were used. Each player used the same? How the variability was controlled if not?

Thank you for this comment. We added information that 21 GPS devices were used. Also, each player wore the same GPS device in all training sessions and matches in order to avoid inter-unit variability. Please find cited text after your next comment.

Lines 145-146: add the reliability and validity levels for each of the main outcomes.

Thank you for this suggestion. We added the reliability and validity data for used GPS system, and text now reads: “Data on the RP of the players were collected by 21 GPS devices (Vector S7, Catapult, Catapult Sports Ltd, Melbourne) with a sampling frequency of 10 Hz. The use of this tracking system has appeared in previous researches [23,24]. Such device was already investigated for metrics, and was found to be appropriately reliable and valid in sport settings (i.e., less than 1% measurement error, and 80% of common variance with running speed measured by timing gates) [19,25,26]. Each player wore the same GPS device in all training sessions and matches in order to avoid inter-unit variability.”

Lines 166-168: add the normality and homogeneity levels of each of the outcomes

Thank you for this suggestion, we added information accordingly. Text now reads: “Homogeneity was checked by the Levene’s test. The normality of the distributions was confirmed by the Kolmogorov–Smirnov test, and data are presented as means ± standard deviations. All variables were normally distributed (all K-S p > 0.05), except TDr-MD4, LIDr-MD4 and HIDr-MD4 (K-S: p < 0.05, 0.05 and 0.01, respectively). Therefore, these three variables were excluded from analysis in Discussion”

Line 169: the repeated measures were made by each player? How did the authors manage the missings?

Yes, repeated measures were made by each player. As we wrote in Design, we included in analysis only players who played a whole match and participated in all training sessions in the week before each match were included in the study. We did this exactly because of missing data issue. In the end, it reduced number of observations to 87, what was already emphasized as limitation of our study. Please find it here: “The main limitation of this study is the fact that only one team was observed. However, this is a very common obstacle in studies involving professional and elite players [15,38]. In addition, we included players only if they participated in full training sessions and the corresponding following matches. Finally, for a more comprehensive understanding of RTL, measures of internal load should be included in future studies.”

RESULTS 

Figures: add the units of measurement in the y-axis

Amended accordingly. Thank you.  

The statistical report in the text must describe the percentage of differences in the pairwise comparisons.

Thank you very much for this comment. We added values for all training/match ratios in Results section where we presented descriptive statistics. Basically, if multiplied by 100, such values describe the percentage of differences between RP in training matches.

DISCUSSION

A sub-section of future research and practical implications should be added at the bottom of the section.

Thank you very much for this suggestion. We systematically rewrote Conclusion and added new subheading ‘Practical implications and future research’. Text now reads:

“4.4. Practical implications and future research

  • Players should be exposed to a minimum of 75–80% of the high-intensity run-ning normally characterizing a match in the middle of the week (i.e., on MD-3 in our study).
  • When utilizing small-sided games which do not require running at high speeds, coaches should include specific running drills that entail high-intensity running (e.g., high-speed running and sprinting) in the training sessions.
  • Training methodology that utilizes a “high-volume and low-intensity” training session the second day after the match (i.e., on MD-5) may positively impact success in soccer.
  • After a heavy training session in the middle of the week, coaches should re-duce the volume of the training on the two days before the match.
  • For a more comprehensive understanding of RTL, measures of internal load should be included in future studies.

  1. Conclusions

The results from our study demonstrate the influence of RTL within weekly training sessions on success in soccer. Specifically, we found that a final match result in soccer may be influenced by greater RTL of performances that determine the train-ing intensity (i.e., high-intensity running) three days before the match.

Additionally, this study indicated that greater RTL of variables that determine the training volume (i.e., total and low-intensity running, accelerations and decelerations) two days after the match (i.e., on MD-5) may positively reflect on a match final result.

Finally, we found that a decrease in RTL variables that determine the training volume one and two days before a match increases the possibility of winning the match. These findings indicate that such approach will maximize players’ performance on match day and, at the same time, reinforce previous knowledge about the effectiveness of tapering strategies in soccer.”

Reviewer 2 Report

The present study is of interest to examine relative training load and match outcome in soccer players. 

The aim of this study were to: 

1) describe the daily training/match ratios of different running performance (RP) variables in order to analyze intra-week variations of relative training load (RTL);

2) identify associations between training/match ratios and match outcome (win, draw, and loss) in order to investigate the possible influence of RTL on success in soccer.

As the main conclusion of this work, the authors found that a match final result in soccer may be influenced by greater relative training load (RTL) of performances that determine the training intensity (i.e., high-intensity running) three days before the match.

Despite the interesting work, I strongly suggest following the comments to improve the quality of the manuscript. The manuscript is generally well written.    

Abstract

1. I suggest adding 95% IC to your results.
2. I suggest to re-write your conclusions based on the main results of the present study. It is no clear enough the main findings/conclusions of this work.

Introduction

3. L 73-75. It is not clear in the introduction section which values for high-intensity running are considered as "good references" for soccer players, to be more trained during the in-season weekly training practice. Authors should provide this information, with respective references, in this section. 

Materials and Methods

4. L 101. Please provide the reference code of the ethical approval. 

5. L 110-113. Please provide and reinforce this sentence with references. 

6. L 115-116. Authors should indicate % missing cases and/or dropouts from the initial selection. 

7.  L 120-137. I suggest the authors to present a flowchart/figure to better facilitate the reading.  

8. L 166-177. Statistical procedures might need to be discussed using a within-subjects approach since basic group comparisons were performed.

- How was this comparison attempted? Did authors pool data for the comparison at the group level? How many data points were paired?
- Can ANOVA for repeated measures or One-way analyses test be used (or be appropriate) for such data set?
- Given the high intra-individual variability, a within-subjects approach (recommended for small samples; Linear mixed model [lmm] and generalized linear mixed model [glmm] analysis) might be appropriate (please see some recent works: https://pubmed.ncbi.nlm.nih.gov/31527865/ , https://pubmed.ncbi.nlm.nih.gov/33672683/; https://www.frontiersin.org/articles/10.3389/fphys.2021.678462/full).

Results 

9. Please provide 95 % IC for all effect sizes. 

10. Table 1. Can authors confirm normality of TDr, LIDr, RDr and HIDr scores? SD values, in some cases, are high.

11. According to the previous comment (10.), please also check for Table 2. 

Author Response

REVIEWER 2

The present study is of interest to examine relative training load and match outcome in soccer players. 

The aim of this study were to: 

1) describe the daily training/match ratios of different running performance (RP) variables in order to analyze intra-week variations of relative training load (RTL);

2) identify associations between training/match ratios and match outcome (win, draw, and loss) in order to investigate the possible influence of RTL on success in soccer.

As the main conclusion of this work, the authors found that a match final result in soccer may be influenced by greater relative training load (RTL) of performances that determine the training intensity (i.e., high-intensity running) three days before the match.

Despite the interesting work, I strongly suggest following the comments to improve the quality of the manuscript. The manuscript is generally well written.    

Thank you for recognizing potential of our work. We tried to follow all your comments and amend it according to your suggestions.

Abstract

1. I suggest adding 95% IC to your results.

Thank you for this suggestion. We added confident intervals in Abstract. Text now reads: “). The results indicated significantly higher HIDr on MD-3 (0.73; 95%CI: 0.57-0.9), higher TDr, LIDr, RDr, ACCr, and DECr on MD-5 (0.71; 0.72, 0.78, 0.65, and 0.66, respectively; 95%CI: 0.63-0.78; 0.68-0.77; 0.48-1-07; 0.55-0.74; 0.57-0.76), and lower TDr, LIDr, ACCr, and DECr on MD-1 (0.35, 0.39, 0.32, and 0.33, respectively; 95%CI: 0.33-0.37, 0.38-0.41, 0.19-0.25, 0.31-0.34, 0.31-0.35) and MD-2 (0.35, 0.4, 0.29, and 0.3, respectively; 95%CI: 0.31-0.38, 0.36-0.43, 0.14-0.23, 0.25-0.33, 0.25-0.34) when a team won in a match played in the same week (F-test: from 6.97 to 34.40; all p < 0.01; medium-to-large effect sizes).”

  1. I suggest to re-write your conclusions based on the main results of the present study. It is no clear enough the main findings/conclusions of this work.

We agree with you, thank you. Conclusion part was re-written to provide main findings/conclusions. Text now reads: These findings demonstrate that final match outcome in soccer was influenced by greater RTL of (i) high-intensity running three days before the match, (ii) total and low-intensity running, accelerations and decelerations five days before the match, and lower RTL of total and low-intensity running, accelerations and decelerations one and two days before the match.”

Introduction

  1. L 73-75. It is not clear in the introduction section which values for high-intensity running are considered as "good references" for soccer players, to be more trained during the in-season weekly training practice. Authors should provide this information, with respective references, in this section. 

Thank you very much for this comment. We totally understand your point here, but unfortunately, to the best of our knowledge there is no study which reported “which values for high-intensity running are considered as "good references" for soccer players to be more trained during the in-season weekly training practice”. Although all previous studies clearly highlighted undertraining for high-intensity running during the in-season weekly training practice of elite soccer teams in different countries, it is still unknown can high-intensity training loads reflect on team’s success in matches. Therefore, results from our study actually present novel findings, reporting that players should be exposed to a minimum of 73% of the high-intensity running normally characterizing a match in the middle of the week (i.e., on MD-3 in our study).

However, to make more clear issues about undertraining of high-intensity running during the weekly training sessions, we amended this part and added that previous studies indicated undertraining of high-intensity running in weekly training sessions in compared to other running performances. Text now reads: “All previously cited studies clearly highlighted undertraining for high-intensity running in compared to the other running performances during the in-season weekly training practice of elite soccer teams in different countries. Although these findings provided valuable information for optimizing the physical conditioning of players, it is still unknown if such lower loads reflect a team’s success in matches.”

 Materials and Methods

  1. L 101. Please provide the reference code of the ethical approval. 

Amended accordingly. Text now reads: “All the players approved the use of training and match data for the purpose of the present study by written consent. The investigation was approved by the Ethical Board of the Faculty of Kinesiology, University of Split (approval number: 2181-205-02-05-19-0020, 1st September 2019)”

  1. L 110-113. Please provide and reinforce this sentence with references. 

Thank you for this suggestion. As we were aware that contextual variables can affect running performance, we excluded some matches that really had significant impact on running performance. To reinforce this issue, we added two references that reported strong influence of contextual variables (i.e., situational and environmental factors) on running performance: https://doi.org/10.3390/ijerph18105175; https://doi.org/10.1371/journal.pone.0247771. Text now reads: “Due the well-known influence of situational and environmental factors on running performance in soccer [23,24], we also did not analyze matches that included red cards or specific outputs (e.g., bad weather, bad pitch, matches against teams that mathematically assured title or relegation) to minimize the influence of those contextual variables on RP.”

  1. L 115-116. Authors should indicate % missing cases and/or dropouts from the initial selection. 

Thank you for this comment. In general, from 36 matches that were played in season, we included in analysis only 14. This was done intentionally due to different reasons, as explained in Method section.  Also, we were “cutting” our sample until we have achieved to not have missing cases any more, what was done due the methodological reasons.

However, we re-wrote this part to provide better insight into the dropouts from the initial selection. Text now reads: “These criteria reduced number of the analyzed matches from 36 (i.e., which is maxi-mum number of matches in observed competition) to 14. Consequently, only training sessions that preceded analyzed matches were observed (n = 67), resulting in 87 observations which were used as cases for this study.”

  1. L 120-137. I suggest the authors to present a flowchart/figure to better facilitate the reading.  

Thank you very much for this suggestion. We added Figure 1 to better facilitate the reading.

  1. L 166-177. Statistical procedures might need to be discussed using a within-subjects approach since basic group comparisons were performed.

- How was this comparison attempted? Did authors pool data for the comparison at the group level? How many data points were paired?
- Can ANOVA for repeated measures or One-way analyses test be used (or be appropriate) for such data set?
- Given the high intra-individual variability, a within-subjects approach (recommended for small samples; Linear mixed model [lmm] and generalized linear mixed model [glmm] analysis) might be appropriate (please see some recent works: https://pubmed.ncbi.nlm.nih.gov/31527865/ , https://pubmed.ncbi.nlm.nih.gov/33672683/; https://www.frontiersin.org/articles/10.3389/fphys.2021.678462/full).

Thank you very much for this comment. We would like to like explain our reasons for using ANOVA. First, we agree that Linear mixed model and Generalized linear mixed model may be appropriate when data are hierarchical structured, what is typical for investigations when sample includes more than one team. However, since our work included only one team, units of analysis (individual match observations) are not multiply clustered. This reduces high intra-individual variability. On the other side, possible variability may still exist because of the players who play at different playing positions. This is well-known fact in the soccer matches, but running performances in single training session do not differ too much irrespective of different playing positions. Please have look results of these paper: 10.1371/journal.pone.0209393, 10.26773/smj.210202. This reduces high intra-individual variability as well.

To be more specific, we will provide detail explanation based on our practical experience (i.e., since we were included in training process of observed players). Basically, players usually have similar running performance in single training session because in almost all parts of the training players participate in similar kind of the work (warm up, te-ta drills, small sided games) that are not position-specific. Moreover, if they participate in position specific drills (for example playing the game in the training), it typically lasts only 10-15 minutes, what corresponds to maximal 15% of the full training session time – what obviously is not enough to provoke differences in running performance. In contrast, when playing the game, players have all 90 minutes different roles, what significantly influence their running performance. Therefore, when analysing data from the matches, linear mixed model is necessary. Since we analysed training loads and since we highly reduced our sample to not have missing data, we thought that ANOVA may be appropriate for analysing our sample.

Results 

  1. Please provide 95 % IC for all effect sizes. 

Thank you for this suggestion. Confidence intervals for all effect sizes were added to Tables 1 and 2.

  1. Table 1. Can authors confirm normality of TDr, LIDr, RDr and HIDr scores? SD values, in some cases, are high.

Thank you for this comment. We added information about normality of distributions according to your suggestion. Text now reads: “The normality of the distributions was confirmed by the Kolmogorov–Smirnov test, and data are presented as means ± standard deviations. All variables were normally distributed (all K-S p > 0.05), except TDr-MD4, LIDr-MD4 and HIDr-MD4 (K-S: p < 0.05, 0.05 and 0.01, respectively). Therefore, these three variables were excluded from analysis in Discussion.

  1. According to the previous comment (10.), please also check for Table 2. 

All variables for accelerations and deceleration that are presented are normally distributed (all K-S p > 0.05). Please look at K-S calculation below:

Variable

Max D

K-S

p

RELTotalDISTANCEmd1

0,062931

p > .20

RELTotalDISTANCEmd2

0,070983

p > .20

RELTotalDISTANCEmd3

0,111860

p > .20

RELTotalDISTANCEmd4

0,156422

p < ,05

RELTotalDISTANCEmd5

0,107762

p > .20

RELZ12md1

0,091541

p > .20

RELZ12md2

0,073642

p > .20

RELZ12md3

0,065288

p > .20

RELZ12md4

0,163063

p < ,05

RELZ12md5

0,180332

p > .20

RELZ3md1

0,083403

p > .20

RELZ3md2

0,070488

p > .20

RELZ3md3

0,106099

p > .20

RELZ3md4

0,129027

p < ,15

RELZ3md5

0,195545

p < ,20

RELZ45md1

0,070597

p > .20

RELZ45md2

0,142422

p < ,10

RELZ45md3

0,141525

p < ,10

RELZ45md4

0,221229

p < ,01

RELZ45md5

0,207917

p < ,15

RELTotACCmd1

0,055868

p > .20

RELTotACCmd2

0,073160

p > .20

RELTotACCmd3

0,083229

p > .20

RELTotACCmd4

0,115420

p > .20

RELTotACCmd5

0,096199

p > .20

RELTotDECEmd1

0,078546

p > .20

RELTotDECEmd2

0,076817

p > .20

RELTotDECEmd3

0,090600

p > .20

RELTotDECEmd4

0,118020

p > .20

RELTotDECEmd5

0,088554

p > .20

RELHighACCmd1

0,139898

p < ,10

RELHighACCmd2

0,119686

p < ,20

RELHighACCmd3

0,140808

p < ,15

RELHighACCmd4

0,143397

p < ,10

RELHighACCmd5

0,194067

p < ,20

RELHighDECEmd1

0,128835

p < ,15

RELHighDECEmd2

0,140069

p < ,10

RELHighDECEmd3

0,120749

p > .20

RELHighDECEmd4

0,111972

p > .20

RELHighDECEmd5

0,119076

p > .20

Round 2

Reviewer 1 Report

The document was improved based on the comments. I would like to endorse the acceptance.

Author Response

Thank you for your support. Additional changes are done according to Editor's comments

Reviewer 2 Report

I am happy with the current version of the manuscript.

The authors did a good job on reviewing the manuscript.

Author Response

Thank you for recognizing the quality of our work. Additional changes are done according to Editor's comments.